# Embarrassingly Parallel Independent Training of Multi-Layer Perceptrons with Heterogeneous Architectures

Felipe C. Farias [1,*,†], Teresa B. Ludermir [1,†] and Carmelo J. A. Bastos-Filho [2,†]

1 Centro de Informática, UFPE, Recife 50740-560, PE, Brazil
2 University of Pernambuco–Ecomp, Recife 50720-001, PE, Brazil
* Correspondence: felipefariax@gmail.com
† These authors contributed equally to this work.

**Abstract:** In this paper we propose a procedure to enable the training of several independent Multilayer Perceptron Neural Networks with a different number of neurons and activation functions in parallel (ParallelMLPs) by exploring the *principle of locality* and parallelization capabilities of modern CPUs and GPUs. The core idea of this technique is to represent several sub-networks as a single large network and use a Modified Matrix Multiplication that replaces an ordinal matrix multiplication with two simple matrix operations that allow separate and independent paths for gradient flowing. We have assessed our algorithm in simulated datasets varying the number of samples, features and batches using 10,000 different models as well as in the MNIST dataset. We achieved a training speedup from 1 to 4 orders of magnitude if compared to the sequential approach. The code is available online.

**Keywords:** neural networks; parallelization; scatter add; gpu; supervised learning

## 1. Introduction

Machine Learning models are used to solve problems in several different areas. The techniques have several knobs that must be chosen before the training procedure to create the model. The choice of these values, known as hyper-parameters, is a highly complex task and directly impacts the model performance. It depends on the user experience with the technique and knowledge about the data itself. Also, it is generally difficult for non-experts in modelling and the problem to be solved due to a lack of knowledge of the specific problem. The user usually has to perform several experiments with different hyper-parameter values to maximize the model's performance to find the best set of hyper-parameters. Depending on the size of the model and the data, this can be very challenging.

The hyper-parameter search can be performed manually or using search algorithms. The manual hyper-parameter search can be done by simply testing different sets of hyper-parameters. One of the most common approaches is to use a grid search. The grid-search process assesses the models by testing all the possible combinations given a set of hyper-parameters and their values. Search algorithms can also be applied by using knowledge of previous runs to test promising hyper-parameter space during the next iteration, such as evolutionary and swarm algorithms, Gaussian processes [1]. In this paper, we focus on the hyper-parameter search for Artificial Neural Networks since it is an effective and flexible technique due to its universal approximator capabilities [2] applied to tabular datasets since it is a prevalent type of data used in the real world by several companies.

The best set of hyper-parameters could be chosen if the technique could be assessed using all the possible values for each variable. However, this is usually prohibitive due to the model training time. Two paths can be taken to accelerate the process: (i) reducing the training time of each model or (ii) assessing the model in the most promising points of the hyper-parameter space. The second point can be challenging because it is hard to perform a thorough search using a few points.

GPUs are ubiquitous when training Neural Networks, especially Deep Neural Networks. Given the computational capacity that we have today, we can leverage the parallelization power of CPUs that contain several cores and threads and GPUs with their CUDA [3] cores to minimize the training time. The parallelization is commonly applied during matrix multiplication procedures, increasing the overall speed of the training process. However, if the data and the model that is being used do not produce big matrices, GPUs can decrease the training time if compared to CPUs due to the cost of CPU-GPU memory transfers [4]. Also, the GPUs might not saturate their usage when using small operations. If we aggregate several MLPs as a single network with independent sub-networks, the matrices become more extensive, and we can increase the GPU usage, consequentially increasing the training speed. In this paper, we designed a single MLP architecture that takes advantage of caching mechanisms to increase speed in CPUs or GPUs environment, called ParallelMLPs from now on, that contains several independent internal MLPs and allow us to leverage the parallelization power of CPUs and GPUs to train individual and independent Neural Networks, containing different architectures (number of hidden neurons and activation functions) simultaneously. This is an embarrassingly parallel alternative to train MLPs as the internal subnetworks' results and computations are performed independently of each other.

## 2. Background

In this section, we present important aspects of hyper-parameter search, computer organization, training speed of machine learning models.

### 2.1. Hyper-Parameter Search

Hyper-parameters can be defined as the general configurations to create the models and control the learning process. They might be discrete, such as the (i) number of layers, (ii) number of neurons, (iii) activation functions, or continuous such as (i) learning rate and (ii) weight regularization penalty, among others. There are several ways to perform a hyper-parameter search. The most common and simple one is the Manual Search. The user can run a grid search to try all the possible combinations given a set of possibilities for every hyper-parameter being searched. In [5], the authors claim that performing a Random Search is more efficient than running a grid search. We believe that the random search is more critical for continuous values since it is more difficult to find the correct answer for that specific hyper-parameter. Nevertheless, a grid search might be the complete assessment of the best discrete hyper-parameter settings for the discrete ones. More knowledge regarding the function space leads to a better choice of the optimized point. In other words, if more architectures are tested, more information is gained about the hyper-parameter that maximizes the performance on a given task.

### 2.2. Computer Organization and Model Training Speed

Several factors can impact the computational speed of an algorithm. Consequentially, there are several ways to improve the efficiency of an algorithm. One of the most important is through the Big-O analysis [6]. However, once the algorithm is optimized regarding Big-O, usually hardware efficiency is related to the algorithm speed. The computer architecture is made up of several components. The Von Neumann model [7] describes the architecture of digital computers. In this architecture, the computer must have (i) a Central Processing Unit (CPU) that controls everything, containing an Arithmetic and Logic Unit (ALU), a program counter (PC), and processor registers (ii) primary memory to store data and instructions, external mass storage, and input/output mechanisms. All the communication happens in a shared bus. This shared bus can be a bottleneck as it can limit the throughput between CPU and memory. Most modern computers operate with multi-core processors containing several cache levels between the CPU and RAM [8] to mitigate this bottleneck. Algorithms aware of memory hierarchy management and computer organization can have severe speedups during execution [9].

The ParallelMLPs training algorithm was designed to mainly take advantage of *principle of locality* [10]. When using several models as a single model (i.e., fusing several matrices), we store this data contiguously, which can be very useful for caching mechanisms to exploit the spatial locality, pre-fetching from memory the input data and model parameters' neighbourhood. Since we are presenting a specific batch for several models simultaneously, the temporal locality can also be exploited once this data might be retained for a certain period. The instructions can also be cached efficiently since, in big matrix multiplication, the instructions will be repeated several times.

The CPU and GPU processors are an order of magnitudes faster than memory transfer operations (RAM or VRAM). When training sequentially, the processors can waste several cycles waiting for specific operations such as batch creation (RAM access, CPU to GPU transfers, CUDA global memory to CUDA shared memory). When training 10,000 models sequentially for 10 epochs in 1000 samples, we will have to randomly access memory (bad caching locality properties) to create the batches $10,000 * 10 * 1000 = 100,000,000$ times. Several cache misses can happen during batch fetching. When we use ParallelMLPs, we increase the chance of cache hit since we are using the current batch on 10,000 models simultaneously. Probably the batch is in the cache closest to the processor during the tensor operations. At the same time, we decrease the number of memory access in case of cache misses to $10 * 1000 = 10,000$ at maximum. Of course, everything described here depends on the computer it is being applied to due to different memory layouts and the number of cores.

### 2.3. Multi-Layer Perceptron

The Multi-Layer Perceptron (MLP) is a prevalent and powerful machine learning model [11]. There are some challenges to training it since the architecture definition is not an easy task and is very dependent on the user experience. The training process achieves the parameter adjustment of an MLP. It can be divided into two phases: forward and backward. In the forward phase, we project the input space into a hidden/latent space, usually applying a non-linear function to a weighted sum of the inputs, projecting this latent space into the output space again to have the predictions. These predictions are compared against the targets that the model is trying to learn through a loss function that measures how good the models' predictions are. In the backward phase, we take the partial derivatives of each parameter w.r.t. the error to propagate it toward the input, updating the parameters accordingly and hopefully decreasing the prediction errors in the following forward phase.

### 3. Materials and Methods

In order to facilitate the methodology explanation, we will write the tensors in capital letters. Their respective dimensions can be presented as subscripts such as $W_{[3,4]}$ meaning a two-dimensional tensor with three rows and four columns.

An example of a MLP architecture with 4 inputs, 3 hidden neurons and 2 outputs $(4 - 3 - 2)$ can be seen in Figure 1. The weights are also explicit with notation *wij* meaning a connection from neuron *j* in the current layer to neuron *i* in the next layer. The first weight matrix with shape $[3, 4]$ projects the input representation (4 dimensions) into the hidden representation (3 dimensions). In contrast, the second matrix with shape $[2, 3]$ projects the hidden representation (3 dimensions) into the output representation (2 dimensions).

The training procedure of an MLP is usually performed by the Backpropagation algorithm [12]. It is divided into two phases. We perform several non-linear projections in the forward phase until we reach the final representation in the backward phase. Then, we can calculate the error using a loss function to backpropagate them using the gradient vector of the error w.r.t. the parameters to update the parameters of the network in order to minimize its error.

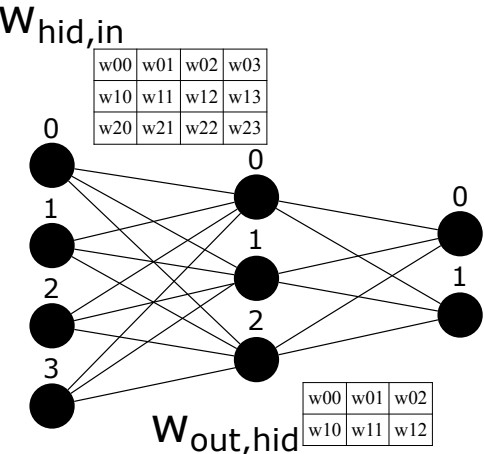

**Figure 1.** Simple MLP with 4 inputs, 3 hidden neurons, and 2 outputs and its weight matrices.

The forward calculation can be described as two consecutive non-linear projections of the type $H = X \times W_1^T$. and $Y = H \times W_2^T$. We need two different weight matrices $w_1$ with shape $[3, 4]$ and $w_2$ with shape $[2, 3]$. We also would want two bias vectors, but we will not use them to facilitate the explanations and figures.

The forward phase in our example can be visualized as a three-step procedure.

1.   Input to hidden matrix multiplication, $H_{batch,hid} = X_{batch,in} \times W_{hid,in}^T$.
2.   Hidden activation function application, $H'_{batch,hid} = \sigma(H_{batch,hid})$.
3.   Hidden activated to output matrix multiplication, $Y_{batch,out} = H'_{batch,hid} \times W_{out,hid}^T$.

To train the model, we need to apply a loss function to compare the numbers in step 3 against the targets which the model is trying to learn. After the loss calculation, the gradient of each parameter w.r.t. the loss is estimated and used to update the parameters of the network.

Since MLP architectures usually do not saturate GPUs, we can change the architecture to create MLPs that share the same matrix instance but have their own independent set of parameters. It allows them to train in parallel. Suppose we want to train two MLPs in the same dataset with architectures $MLP_1 = 4 - 1 - 2$ and $MLP_2 = 4 - 2 - 2$. To leverage the GPU parallelization, we can fuse both architectures as a single architecture (ParallelMLPs), in the form $4 - 3 - 4$. In Figure 2 we can see both internal networks represented as a single architecture (ParallelMLPs) with different colours. The number of inputs does not change; the number of hidden neurons is summed, while the number of output neurons is multiplied by the number of independent MLPs we want to train. The layout of ParallelMLP was designed to take advantage of temporal and spatial locality principles in caching [10] and overall parallelization mechanisms.

In order to maintain the internal MLPs independent of each other, we cannot follow the same steps previously described because, in that case, we would mix the gradients of all the internal models during the backpropagation. The matrix multiplication can be understood as two consecutive operations: (i) element-wise vector multiplication (rows $*$ columns) and (ii) reduced sum of the previous vectors. We need to change how we perform the matrix multiplication in step 3. The main idea is to divide the matrix multiplication into two procedures: (i) matrix element-wise multiplication and (ii) summation. However, the summation needs to be carefully designed such that we do not reduce-sum the entire vector but different portions of the axis. We can use broadcast techniques implemented in every tensor library, and a special case of summation called Scatter Add to make these two procedures memory efficient. We will refer to this procedure as Modified Matrix Multiplication (M3) [13].

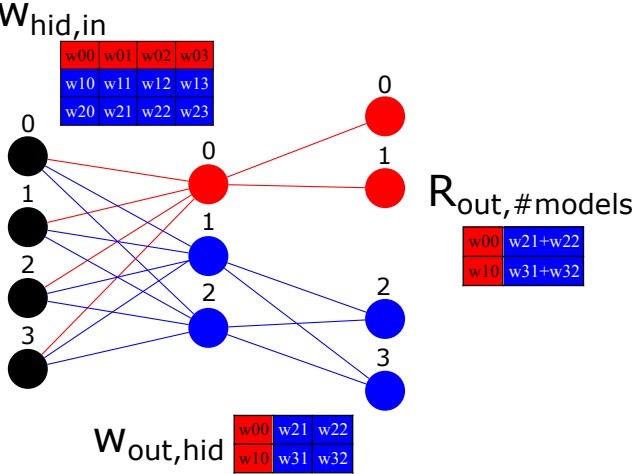

**Figure 2.** Two independent MLPs are represented as a single MLP with four inputs, 1 and 2 hidden neurons, and two outputs. The weight matrices are also highlighted to understand the parameter mapping from the single MLP architecture to its two internal MLP architectures. The red colour is related to the $4 - 1 - 2$ MLP, while the blue colour is related to the $4 - 2 - 2$ MLP. $R_{out,\#models}$ contain the result after applying the Scatter Add operation.

This M3 procedure might be useful to handle sparse NN. Most of the time, sparsity is treated with masking. Masking tends to be a waste of resources since the original amount of floating point calculations are still being done, and additional masking floating point operations are being added to the process.

The Scatter Add ($\phi(D, S, I)$) operation takes a dimension $D$ to apply the operation and two tensors as inputs. The source tensor $S$ contains the numbers that we want to sum up and the indices tensor $I$, which informs which elements in the source tensor must be summed and stored in the result tensor. When applied to two-dimensional tensors as $\phi(1, S, I)$, the result tensor $R$ (initialized as zeros) can be calculated as:

- $R[I[i, j], j] = R[I[i, j], j] + S[i, j]$ if dimension = 0
- $R[i, I[i, j]] = R[i, I[i, j]] + S[i, j]$ if dimension = 1

A very simple example of the scatter add operation would be when:

- $D = 1$,
- $S_{[1,6]} = [[1, 2, 3, 4, 5, 6]]$,
- $I_{[1,6]} = [[0, 1, 1, 2, 2, 2]]$,
- $R_{[1,3]} = \phi(1, S, I)_{[1,3]} = [[1, 5, 15]]$,

with $I$ informing how to accumulate values in the destination tensor $R$, with the first element (0) accumulating only the first element of $S$, the second destination element (1) accumulating the second and third values of $S$, and the latest element (2) accumulating the fourth, fifth and sixth element of $S$. This operation is implemented in parallel in any popular GPU tensor library. In our architecture represented in Figure 2, in order to have the two separated outputs, one for each internal MLP, we would have to sum across the lines using a matrix $I$ as follows:

$$I_{[2,3]} = \begin{bmatrix} 0 & 1 & 1 \\ 0 & 1 & 1 \end{bmatrix}$$

It would generate an output matrix $[2, 2]$ where each line is related to each internal/individual MLP.

The Scatter Add operation is responsible for keeping the gradients after the loss function application not mixed during the backpropagation algorithm, allowing us to train thousands of MLPs simultaneously in an independent manner.

To summarize, the steps to perform the parallel training of independent MLPs are:

1.  Input to hidden matrix multiplication, $H_{batch,hid} = X_{batch,in} \times W_{hid,in}^T$.
2.  Hidden activation function application, $H'_{batch,hid} = \sigma(H_{batch,hid})$.
3.  Hidden activated element-wise multiplication with multi-output projection matrix, $S_{batch,out,hid} = H'_{batch,1,hid} \odot W_{1,out,hid}$ (broadcasted element-wise multiplication)
4.  Finally, the Scatter Add operation to construct the independent outputs $Y_{batch,model,out} = \phi(1, S_{batch,out,hid}, I_{batch,out,hid})$

After that, a loss function is applied to calculate the gradients and update all the internal MLPs parameters independently.

We can go further and use not only a single activation function that should be applied to all the internal MLPs, but several of them by repeating the original number of architectures as many times as we have for activation functions. Since this is often not enough to use all the GPU resources, one can also have repetitions of the same architecture and activation function. In order to use several activation functions, the last step must be modified such that we can apply different activation functions to different portions of the matrix *O*. This can be done using a tensor split operation, applying each activation function iterativelly in different continuous portions of the representations, and finally concatenating the activated representations again.

The Python pseudocode of our proposal is described in Pseudocode 1. The complete code is available at Github (https://github.com/fariasfc/parallel-mlps).

Pseudocode 1: High-level description of the ParallelMLPs operations.

```python
import torch
class ParallelMLPs:
    def __init__(self, in_features, out_features, min_neurons,
        max_neurons, step_neurons, activations, repetitions):
        neurons_structure = torch.arange(min_neurons, max_neurons +
            1, step).tolist()
        num_activations = len(activations)
        num_parallel_mlps = len(neurons_structure) *
            num_activations * repetitions
        total_hidden_neurons = neurons_structure.sum() *
            num_activations * repetitions

        self.activations = activations
        self.hidden_layer = Linear(in_features,
            total_hidden_neurons)
        self.hidden_neuron_model_id = # tensor mapping a hidden
            neuron to the subnetwork id

        # Equivalent to the output_layer
        self.weight = Parameter(
            torch.Tensor(out_features, total_hidden_neurons)
        )
        self.bias = Parameter(
            torch.Tensor(num_parallel_mlps, self.out_features)
        )

    def forward(self, x: Tensor) -> Tensor:
        batch_size = x.shape[0]
        x = self.hidden_layer(x)  # [batch_size,
            total_hidden_neurons]
```

```
24          x = self.apply_activations(x)  # [batch_size,
         ↪  total_hidden_neurons]

25

26          x = (
27              x[:, :, None] * self.weight.T[None, :, :]
28          )  # [batch_size, total_hidden_neurons, 1] * [1,
         ↪  total_hidden_neurons, out_features] = [batch_size,
         ↪  total_hidden_neurons, out_features]

29

30          adjusted_out = (
31              torch.zeros(
32                  batch_size, self.num_unique_models,
                 ↪  self.out_features, device=x.device
33              ).scatter_add_(
34                  1,
35                  self.hidden_neuron__model_id[None, :, None].expand(
36                      batch_size, -1, self.out_features
37                  ),  # [batch_size, total_hidden_neurons,
                 ↪  out_features], expand does not consume extra
                 ↪  memory.
38                  x,
39              )
40          ) + self.bias[None, :, :]

41

42          # [batch_size, num_unique_models, out_features]
43          return adjusted_out
```

### 3.1. Experiments

Simulations were performed in order to compare the speed of the Parallel training against the Sequential approach.

### 3.1.1. Computing Environment

A Machine with 16 GB RAM, 11 GB NVIDIA GTX 1080 Ti, and an I7-8700K CPU @ 3.7 GHz containing 12 threads were used to perform the simulations. All the code was written using PyTorch [14].

### 3.1.2. Model Architectures

We have created architectures starting with one neuron at the hidden layer until 100, resulting in 100 different possibilities. For each architecture, we have assessed ten different activation functions: *Identity, Sigmoid, Tanh, ReLU, ELU, SeLU, GeLU, LeakyReLU, Hardshrink, Mish*. It increases the number of independent architectures to $100 * 10 = 1000$. We have repeated each architecture 10 times, totalling $1000 * 10 = 10,000$ models. It is worth mentioning that this design is not limited to these circumstances. One could create arbitrary MLP architectures such as 3 different networks with 3, 19, and 200 hidden neurons and still would be able to leverage the speedup from ParallelMLPs.

### 3.1.3. Datasets

We have created controlled training datasets with 100, 1000, and 10,000 samples. With 5, 10, 50, and 100 features. Giving a combination of 12 different datasets. For all the simulations, 12 epochs were used, ignoring the first two epochs as a warm-up period. For the MNIST dataset, we kept the same amount of model architectures, and investigated only the GPU approach with batch size of 64 due to memory restrictions as the number of features increased to 784.

3.1.4. Training Details

All the samples are stored in GPU at the beginning of the process to not waste much time of GPU-CPU transfers. It favours the Sequential processing speed more than the Parallel since the former would have 10,000 more CPU-GPU transfers throughout the entire experiment.

The data is used only as train splits because this phase is where the gradients are calculated, and the two operations of forward and backward are used. Therefore, the training split processing is much more expensive than validation and test splits.

## 4. Results

The following tables contain the average training time of 10 training epochs (forward and backward steps, no validation, and no test loops) when varying the number of samples (columns) and the number of features (rows) for strategies: (i) Parallel (using ParallelMLPs), (ii) Sequential (training one model at the time), and (iii) the percentage of ParallelMLPs training times against the Sequential strategy (Parallel/Sequential). The CPU and GPU results are presented in Table 1 and Table 2, respectively.

**Table 1.** Average of 10 epochs to train 10,000 models using a CPU.

| | **Number of Samples** | | | | | | | | |
| | **100** | | | **1000** | | | **10,000** | | |
| | **Batch Size** | | | | | | | | |
| | **32** | **128** | **256** | **32** | **128** | **256** | **32** | **128** | **256** |
| **Features** | **Parallel (Seconds)** | | | | | | | | |
| **5** | 0.525 | 0.463 | 0.472 | 5.248 | 4.709 | 4.717 | 52.737 | 46.929 | 47.133 |
| **10** | 0.539 | 0.466 | 0.475 | 5.338 | 4.722 | 4.742 | 53.351 | 47.089 | 47.153 |
| **50** | 0.658 | 0.505 | 0.501 | 6.144 | 4.943 | 4.887 | 62.664 | 49.791 | 48.85 |
| **100** | 0.809 | 0.547 | 0.551 | 7.373 | 5.366 | 5.1 | 74.661 | 53.09 | 50.965 |
| | **Sequential (Seconds)** | | | | | | | | |
| **5** | 13.437 | 6.097 | 5.994 | 112.054 | 56.999 | 48.852 | 1097.599 | 564.428 | 483.278 |
| **10** | 13.36 | 6.115 | 6.01 | 111.84 | 57.094 | 49.015 | 1097.503 | 564.701 | 484.91 |
| **50** | 13.884 | 6.571 | 6.467 | 116.303 | 58.993 | 49.546 | 1134.955 | 583.468 | 491.269 |
| **100** | 14.283 | 6.671 | 6.592 | 120.297 | 59.259 | 50.371 | 1179.405 | 586.267 | 493.744 |
| | **Parallel/Sequential (%)** | | | | | | | | |
| **5** | 3.91 | 7.59 | 7.88 | 4.684 | 8.262 | 9.656 | 4.805 | 8.314 | 9.753 |
| **10** | 4.038 | 7.625 | 7.905 | 4.773 | 8.27 | 9.674 | 4.861 | 8.339 | 9.724 |
| **50** | 4.739 | 7.69 | 7.753 | 5.282 | 8.379 | 9.863 | 5.521 | 8.534 | 9.944 |
| **100** | 5.664 | 8.198 | 8.362 | 6.129 | 9.056 | 10.126 | 6.33 | 9.056 | 10.322 |

We also have compared the same architectures being trained on MNIST, containing 60,000 samples with 784 features using 64 as the batch size. We could increase the batch size if we reduce the number of parallel architectures. However, we decided to keep the same architectures for better comparisons. To train 10,000 models for 10 epochs using GPU, we have an average of 118.3 s per epoch using the parallel strategy, while 6348.3 s per epoch using the sequential strategy, the parallel approach corresponds to 1.86% of the time used by the sequential approach.

**Table 2.** Average of 10 epochs to train 10,000 models using a GPU.

| | Number of Samples | | | | | | | | |
| --- | --- | --- | --- | --- | --- | --- | --- | --- | --- |
| | 100 | | | 1000 | | | 10,000 | | |
| | Batch Size | | | | | | | | |
| | 32 | 128 | 256 | 32 | 128 | 256 | 32 | 128 | 256 |
| **Features** | Parallel (Seconds) | | | | | | | | |
| **5** | 0.024 | 0.002 | 0.001 | 0.269 | 0.177 | 0.203 | 2.676 | 1.756 | 2.426 |
| **10** | 0.025 | 0.002 | 0.001 | 0.276 | 0.178 | 0.206 | 2.745 | 1.767 | 2.439 |
| **50** | 0.033 | 0.002 | 0.001 | 0.351 | 0.193 | 0.218 | 3.483 | 1.926 | 2.569 |
| **100** | 0.043 | 0.002 | 0.001 | 0.449 | 0.215 | 0.233 | 4.438 | 2.128 | 2.75 |
| | Sequential (Seconds) | | | | | | | | |
| **5** | 22.911 | 8.646 | 8.511 | 189.411 | 73.503 | 57.02 | 1857.653 | 722.915 | 566.592 |
| **10** | 22.983 | 8.619 | 8.515 | 188.966 | 73.462 | 56.941 | 1859.14 | 722.925 | 568.583 |
| **50** | 23.025 | 8.628 | 8.519 | 189.147 | 73.364 | 57.134 | 1858.07 | 719.359 | 567.847 |
| **100** | 22.993 | 8.581 | 8.503 | 189.015 | 72.849 | 57.129 | 1854.881 | 717.543 | 566.248 |
| | Parallel/Sequential (%) | | | | | | | | |
| **5** | 0.106 | 0.019 | 0.017 | 0.142 | 0.241 | 0.355 | 0.144 | 0.243 | 0.428 |
| **10** | 0.11 | 0.019 | 0.017 | 0.146 | 0.243 | 0.362 | 0.148 | 0.245 | 0.429 |
| **50** | 0.142 | 0.019 | 0.017 | 0.185 | 0.264 | 0.381 | 0.187 | 0.267 | 0.452 |
| **100** | 0.186 | 0.018 | 0.017 | 0.237 | 0.294 | 0.408 | 0.239 | 0.297 | 0.486 |

## 5. Discussion

Suppose one is training for 100 epochs of the previously mentioned 10,000 models in a dataset with 10,000 samples and 100 features with 32 as the batch size. In that case, CPU-Sequential can take more than 32 h ($1179.405 * 100/3600 = 32.76$), while CPU-Parallel only 2 h ($100 * 74.661/3600 = 2.07$) would be necessary to perform the same training. The same case with batch size of 256 samples, we would have approximately 14 h and 1.5 h for CPU-Sequential and CPU-Parallel, respectively. In this case, the CPU-Parallel is 15.8 for the first scenario and 9.3 for the second scenario times faster than CPU-Sequential. The CPU-Parallel experiments took between 3.9% and 10.3% of the CPU-Sequential time, considering all the variations we have performed. As one can see, the CPU speed improves when using larger batch sizes probably to better exploration of the *principle of locality*.

If we analyze the same experiments in GPUs, more than 51 h ($1854.881 * 100/3600 = 51.5$) would be necessary for GPU-Sequential, and 7.4 min when using GPU-ParallelMLPs for the 32 batch experiment and 15.5 h for GPU-Sequential and 4.6 min for the 256 batch size. It gives us a speed improvement on GPU-Parallel of 417.6 and 202.17 times, respectively, when compared to GPU-Sequential. The GPU-Parallel experiments range from 0.017% to 0.486% of the GPU-Sequential time for all the assessed experiments. As one can see, the GPU speed improves when using larger batch sizes probably to better exploration of the *principle of locality* and also a better parallelization in the GPU kernel.

At first glance, the GPU training time should be faster than the CPU training time. However, when we compare GPU-Sequential against the CPU-Sequential, the GPU version is slower than CPU one. This slowness may be explained by the high number of function/kernel calls to perform cheap operations (small matrix multiplications). As the single-core of a CPU is optimized to perform a specific computation very quickly (high clock rate) and the single-core of a GPU is much slower than CPU, it is reasonable that CPUs can take advantage in this scenario. On the other hand, GPUs contain much more cores than

the CPU, and they can run computations in parallel, therefore we can see huge differences as this is often the scenario in which GPUs will outperform CPUs. As we increase the size of matrices to be multiplied in GPU-ParallelMLP (with more architectures or bigger batch sizes), a considerable amount of speed can be delivered compared to CPU-ParallelMLP.

It is essential to mention that the GPU memory consumption of the 10,000 parallel models using 100 features and batch size of 256 (the worst case scenario for our experiments w.r.t. memory allocation) was less than 4.8 GB, meaning that (i) simpler GPUs can be used and still take advantage of our approach, and (ii) is probably possible to improve the speed if using more models in parallel to make a better usage of the GPU memory.

As we can perform a very efficient grid-search in the discrete hyper-parameters space that will define the network architecture, it is much easier for the user to select a suitable model since several number of neurons and activation functions can be trained in parallel, mainly for beginners in the field who have difficulties to guess the best number of neurons and activation function, since the hyper-parameter definition is highly dependent on the user experience. Also, researchers that use any search method that proposes an MLP architecture in a specific problem can now train the models in parallel. The ParalleMLPs can be applied for both classification and regression tasks. We believe this M3 operation can be optimized and lead to even more speed improvements if a specialized CUDA kernel could be written.

## 6. Conclusions

We have demonstrated how to parallelize a straightforward core operation of Neural Networks by carefully choosing alternative operations with a high degree of parallelization in modern processors (CPUs or GPUs).

The ParallelMLPs algorithm described in Section 3 was able to accelerate the training time of several independent MLPs with a different number of architectures and activation functions from 1 to 4 orders of magnitude by simply proposing an efficient memory representation layout that fuses several internal MLPs as a single MLP and using the M3 strategy to perform the matrix projection in an independent way. It allows us to explore better the *principle of locality* and the parallelization in modern processors (CPUs and GPUs). The technique can be helpful in several areas to decrease the training time and increase the number of model assessments. Since we are able to train thousand of networks in a reasonable time, we can investigate the distribution of models for a specific dataset in a large scale. Also, it might be a good way to represent sparse NN.

## 7. Future Works

We believe the ideas proposed in this paper can inspire other researchers to develop parallelization of other necessary core operations/modules such as Convolutions, Attention Mechanisms, and Ensemble fusion of heterogeneous MLPs.

In future works, we would like to investigate if the M3 operation can be used from the second transformation until the last layer to train MLPs with more than one hidden layer since only during the first transformation (from input to the first hidden layer) all the previous neurons are sum-reduced instead of a sparse version of them. We can see an example of this idea into Figure 3.

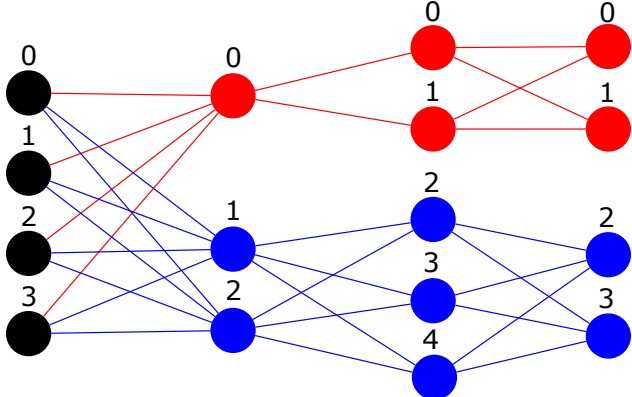

**Figure 3.** Two independent two hidden layers MLPs represented as a single MLP by ParallelMLP. The first with architecture in red being $4 - 1 - 2 - 2$ and the second in blue $4 - 2 - 3 - 2$. The weight matrices were removed to ease the readability of the figure.

An interesting work would be to perform feature selection using ParallelMLPs by repeating the MLP architecture and creating a mask tensor to be applied to the inputs before the first input to hidden projection. We also plan to perform model selection in the large pool of trained MLPs in a specific dataset. Also, we plan to automatize the number of neurons and the number of layers. After we finish the ParallelMLP training, we can (i) remove the output layer or (ii) use the output layer as the new representation of the dataset to be the input of a new series of ParallelMLP training. It is also possible to use the original features concatenated with the previously mentioned outputs like residual connections [15]. After each ParallelMLP training, we can pick the best MLP to create the current layer, continuously increasing the number of layers until no more improvements are perceived. A further investigation is needed to verify if similar ideas could be used for convolutional and pooling layers since they are basic building blocks for several Deep Learning architectures. We also would like to investigate what happens if an architecture containing a backbone representing the input space into a latent space and MLP at the end, such as [16] to perform the classification would be replaced by a parallel layer with several independent set of outputs, but sharing the backbone's parameters. One hypothesis is that the backbone would be regularized by different update signals from a heterogeneous set of MLP heads. This technique can also be similar to a Random Forest [17] of MLPs where during the training phase, the inputs could be masked depending on the individual network to mimic bagging or mask specific features, or even simulate a Random Subspace [18]. Our technique can be used to find the best random initialized model given that [19] was able to found good sub-networks without any training–extending the idea of Lottery Ticket Hypothesis [20]. There is space to parallelize even more hyper-parameters such as batch size, learning rate, weight decay, and initialization strategies. One straightforward way is to use boolean masks to treat each case or hooks to change the gradients directly, but it might not be the most efficient way. A handy extension of our proposition would be to automatically calculate the hyper-parameter space to saturate the GPU.

**Author Contributions:** All authors contributed equally to this work. All authors have read and agreed to the published version of the manuscript.

**Funding:** This research received no external funding.

**Informed Consent Statement:** Not applicable.

**Data Availability Statement:** Not applicable.

**Acknowledgments:** We would like to acknowledge the Centro de Tecnologias Estratégicas do Nordeste (CETENE) for providing computational resources (FACEPE APQ-1864-1.06/12) as well as CNPq and CAPES (Brazilian Research Agencies) for their financial support.

**Conflicts of Interest:** The authors declare no conflict of interest.

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
