# Peer review of "Embarrassingly Parallel Independent Training of Multi-Layer Perceptrons with Heterogeneous Architectures"

_ai, doi:10.3390/ai4010002_

Round 1

Reviewer 1 Report

Authors present a way to parallelise operations that accelerate the training of fully connected neural nets by proposing an alternative way to run matrix multiplication (not considering other operations though, e.g. convolutions).

Given the authors have considered a simple architecture and focusing on a specific issue, the experimental setup appears to be appropriate.

A couple of things:

a) I am not sure why you include the word ‘embarrasingly’ in your title; what does it refer to? I suggest removing it

b) The way you have described your concept is hard to follow; I suggest to add a pseudoalgorithm to describe the steps

c) I think if you were to use a dataset like MNIST (you do not need convolutions) showing how a vanilla way is used to train an MLP with it versus your proposed method, it would make it easier for others to understand the extent to which your method can help to accelerate training

Author Response

REVIEW 1:
Thanks for the review and comments!
> a) I am not sure why you include the word ‘embarrasingly’ in your title; what does it refer to? I suggest removing it
This is a common term in parallel computing. We have described why we call it embarrasingly on lines 53-55.

> b) The way you have described your concept is hard to follow; I suggest to add a pseudoalgorithm to describe the steps
We have added the pseudocode in the paper as well as the github repository www.github.com/fariasfc/parallel-mlps.

> c) I think if you were to use a dataset like MNIST (you do not need convolutions) showing how a vanilla way is used to train an MLP with it versus your proposed method, it would make it easier for others to understand the extent to which your method can help to accelerate training
Indeed. We have added a MNIST simulation. New text were added in lines 7-8, 244-247, and 264-270.

Reviewer 2 Report

The present manuscript address the problem of parellizing the processing of a multi-layer neural network. The results, summarized in tables 1/2, show that the author's procedure can actually accelerate the running times for many arquictectures. However, it has to be provided the reason for the large differences between CPU and GPU performance.

Also, the paper abstract and introduction should be improved to describe the issue adequately. And, most important, there must be a change in the explanation of the 4 steps for parallel training, showing explicitly how to implement it.

  1. authors should provide a more detailed description of the implementation of the "4 steps" for the MLP parallelizing briefly explained at the end of page 5. This could be done either by presenting a table with a pseudocode in the text, or through a full-code referred to in the bibliography for a git-hub like repository. 

     2. authors must rewrite the abstract - the first sentence for instance lacks any information about their work - so that a wider audience can understand what the paper is about.         

3. A reasonable justification for the difference between the CPU and GPU performance should also be provided. Perhaps a graphical plot comparing both time cost behavior could help it.  

Although the paper can be improved with that, I believe that their results are sound, and the conclusions are consistent, as one can see in the tables for the times presented in the end the paper.  

Author Response

REVIEW 2:
Thanks for the review and comments!
> 1. authors should provide a more detailed description of the implementation of the "4 steps" for the MLP parallelizing briefly explained at the end of page 5. This could be done either by presenting a table with a pseudocode in the text, or through a full-code referred to in the bibliography for a git-hub like repository. 
We have added the pseudocode in the paper as well as the github repository www.github.com/fariasfc/parallel-mlps.

> 2. authors must rewrite the abstract - the first sentence for instance lacks any information about their work - so that a wider audience can understand what the paper is about.         
We have changed the abstract to directly reflect the paper proposal.

3. A reasonable justification for the difference between the CPU and GPU performance should also be provided. Perhaps a graphical plot comparing both time cost behavior could help it.  
Usually for simple computations that does not have many parallelizeable pieces, the CPU tends to be faster (which is the case of the Sequential approach). We have incremented the explanaitons in lines 291-301 to make it more clear.
